

# A Bayesian approach to optimizing cryopreservation protocols

Sammy Sambu

Nandi Hills, Kenya

## ABSTRACT

Cryopreservation is beset with the challenge of protocol alignment across a wide range of cell types and process variables. By taking a cross-sectional assessment of previously published cryopreservation data (sample means and standard errors) as preliminary meta-data, a decision tree learning analysis (DTLA) was performed to develop an understanding of target survival using optimized pruning methods based on different approaches. Briefly, a clear direction on the decision process for selection of methods was developed with key choices being the cooling rate, plunge temperature on the one hand and biomaterial choice, use of composites (sugars and proteins as additional constituents), loading procedure and cell location in 3D scaffolding on the other. Secondly, using machine learning and generalized approaches via the Naïve Bayes Classification (NBC) method, these metadata were used to develop posterior probabilities for combinatorial approaches that were implicitly recorded in the metadata. These latter results showed that newer protocol choices developed using probability elicitation techniques can unearth improved protocols consistent with multiple unidimensionally-optimized physical protocols. In conclusion, this article proposes the use of DTLA models and subsequently NBC for the improvement of modern cryopreservation techniques through an integrative approach.

## INTRODUCTION

Cryopreservation of cells often results in cell survivals that are lower because of suboptimal process variables. When cryopreservation is performed in biological constructs (including encapsulated cells), the results are often worse due to the differentials in cryoprotectant (CPA) concentration, CPA exposure times and cooling rates. These differences are exaggerated for cells that are farthest from the surface which is typical for large specimens (*Muldrew et al., 2000*). Conversely, for small capsules that are just above the upper limit of the capsule diameter, differentials in temperature and concentration are meagre (*Cui et al., 2002*) i.e., lower size leads to lower barriers to diffusion and heat transfer leading to smaller CPA concentration and temperature (C–T) differentials—for single cells in suspension, the challenge is further reduced as the cell is closer to C–T equilibrium with its surroundings than encapsulated cells. The main challenge for capsules remains the varying CPA concentrations and exposure times for cells encapsulated in them.

Corresponding author
Sammy Sambu,
Sambu@post.harvard.edu

**How to cite this article** Sambu (2015), A Bayesian approach to optimizing cryopreservation protocols. **PeerJ** 3:e1039;
A spherical capsule, for example, will have different cell survivals at each radius (*Sambu et al., 2011*). Furthermore, there are new variables arising from 3D cryopreservation including cell encapsulation densities, cooling rates and the determiners for CPA concentration profiles e.g., initial Concentration, temperature and construct size (*Sambu et al., 2011*). The challenge is to minimize the deleterious effects of these differentials by providing cryoprotection without mass transfer limitations. Therefore, new construct variables arise. Hence, a re-prioritization of process and capsule geometry variables becomes necessary—a DTLA approach can help resolve the tensions in *prioritizing* and then *recursively* optimizing these variables.

In using a decision-tree learning analysis (DTLA) approach, it is necessary to leverage existing principles in cryopreservation. The entrapment of CPAs within the capsule during cell resuspension in encapsulating material is an alternative strategy to the modulation of cell location. However, intracellular CPAs are often toxic and encapsulation is not lossless. Hence, when a potentially cytotoxic CPA such as DMSO is added to the encapsulating material, cell survival is often diminished (*Cui et al., 2002*; *Liu, 2007*). Therefore, the success of the strategy relies on using a benign CPA as the "entrapped CPA" followed by the loading of the potentially malign but fast-diffusing CPA into the construct. For protein-based cryoprotection, mixing the proteins with polar CPAs also has a deleterious effect on the ability of the protein to confer cryoprotection due to the potential denaturation of the proteins prior to cooling (*Meryman, 2007*). Overall, the protein-DMSO interaction is minimized by incorporating the protein in the encapsulating material, and allowing a time-limited diffusion of the malign CPA into the construct prior to cooling. Besides modifying the encapsulation process, slow-cooling procedures should also be adopted to minimize the required concentrations of the potentially malign CPA.

Besides using proteins and intracellular CPAs, synthetic non-penetrating polymers (SNPP) can also provide cryoprotection within the scaffold, thereby bypassing the limitations of diffusion in higher-dimensional cryopreservation. A subclass of these SNPPs are the vitrifying polymers which have been used to encourage extracellular vitrification of the cryopreservative during cooling thereby limiting ice crystal growth (*Gibson et al., 2009*). Vinyl-derived polymers have also been shown to decrease ice crystal size colligatively. Examples include polyethylene glycol (PEG), polyvinyl alcohol (PVA) and hydroxyethyl starch. The key challenge in using these polymers is minimizing large increases in the viscosity of the solution (which will make encapsulation difficult) and minimizing the difficulty in post-thaw extraction of cells from the scaffold.

Shifting the focus from solutes to encapsulates, changing the encapsulation material (the matrix) properties by using polymer composites can be avoided if one entraps lower molecular weight encapsulates. One alternative would be to use low molecular weight polymers. Another would be to use sugars with cryoprotective properties. The mechanism for cryoprotection conferred by sugars is three-fold: complex sugars can inhibit the formation of ice (*Roos & Karel, 1991*; *Shirakashi et al., 2003*); they can replace lost water molecules and can stabilize lipid bilayers from sudden phase transitions during cooling (*Arakawa et al., 1990*; *Rudolph & Crowe, 1985*). They may prevent sudden changes

in the lipid phase and ensuing phase separation through hydrogen bond formation with phospholipid head groups (*Prestrelski, Arakawa & Carpenter, 1993*; *Anchordoguy et al., 1987*). This interaction causes a reduction in the liquid crystal-gel transition temperature and minimizes membrane fusion by separating one layer from another (*Arakawa & Timasheff, 1982*). Sugars also stabilize proteins by encouraging a preferential hydration of the proteins in solution (*Huang et al., 2003*). Granted that complex sugars are non-permeating solutes, these benefits are extracellular unless a delivery mechanism is engineered.

Each of the techniques mentioned can be represented as random decision variables (*factors*) in the abstract; each one taking on a probability state according to a protocol developed collectively and based on categorical field-specific knowledge. In that way, a DTLA will describe the epistemology of the factors and their relationship to a final probability state describing the life/death or living (long-living)/apoptotic (short-living) state of the hypothetical cell. However, the ability to generalize these DTL relationships is often severely reduced unless sophisticated pruning algorithms are used (*Fürnkranz, 1997*) (conversely local results are definitively optimized based on the learning set) hence requiring the use of a more generalized approach taking the final hypothetical cell state as discrete and each influencing factor as conditionally independent (given nodal classification) in the decision-making process as well as in the execution of the protocol.

Once the requirement for broader applicability of a learned model is defined, a more general approach to making predictions on related but distinct data sets is required. In this latter case, the Naïve Bayes Classifier (NBC) presents a more efficient and direct approach to prediction. The assumption of *class-conditional* factor independence shrinks the pre-prediction computational requirements significantly and speeds the overall time-to-decision making. Nonetheless, empirical evidence must be invoked to support a combined approach using both NBC and DTL (*Rubinstein & Hastie, 1997*). Overall, from the decision-maker's view, by combining the content-centric approach of the DTLA and the boundary-focussed NBC, the opportunity cost is low and the decision process is retrospectively holistic and exhaustive.

This paper seeks to demonstrate the promising strategies for successful cryopreservation of cells in 3D scaffolds by using a DTLA approach to develop a heuristic for approaching cryopreservation across many subjects. Given this cryopreservation challenge specifically concerns the opportunities in 3D cryopreservation; success is measured by a cell survival percentage that is either equal to or better than suspension cryopreservation.

# MATERIALS AND METHODS

## Data collection

Data was collected and categorized into the main decision factors namely: dimensionality, cell location, use of a biomaterial, lyoprotection, CPA loading, use of integrins, CPA type, containment, cell line, cooling rate and plunge temperature (11 features) against 153 instances. Where original data was not present, the mean and standard error were used to redraw virtual samples from the original population described by the sample parameters.

The predicted property is the survival (as a percent for DTLA with regression or as a category for classification with NBC i.e., long-surviving or short-surviving)

While there are many studies that could be incorporated, the key requirement was the explicit description of decision factors that were continuous from study to study; additionally, the studies were chosen to be different enough to incorporate diverse decision factors but of sufficient relational value as to allow a successful DTLA and/or NBC analysis without algorithmic or model instability—in this regard, many failures were met when studies failed to have the same number of sufficient match critical factors (though of overall good cryopreservation results) to allow acceptable predictions and classifications.

## Decision tree learning analysis

Analysis was performed using *Rpart* package in the open source program R for recursive partitioning and regression analysis (*Therneau & Atkinson, 2015*). Note, since the technique allows both classification and regression, the cell survival could be represented either as a class (categorical) or regressed as a continuous variable (continuous).

In brief, the collected data (*Sambu et al., 2011*) and meta-data (*Sambu et al., 2011*; *Heng, Yu & Ng, 2004*; *Kashuba Benson, Benson & Critser, 2008*; *Miszta-Lane et al., 2007*) was imported into R, partitioned according to a variable that best describes the data at each given partition level. The model is 5-fold cross-validated (giving >30 samples as test set for each model instantiation) and the model that best describes the data set is selected after evaluation for fidelity to the original data. For each node, sample numbers are provided. Additionally, the *p*-value is provided for each split showing whether the branching can be supported at 95% confidence.

The DTLA process involves the development of a loss function, computation of a residual over which the learner model is fit. To minimize the loss function, a multiplier is chosen such that through the reapplication of the learner on the training set, the loss function successively diminishes until the tree is fully constructed (*Friedman, 2001*). The overall governing equation is provided below:-

**Equation 1: Governing Equation for the DTLA algorithm where *h* is the regression tree fitted onto the negative gradient 147 (middle term) containing the minimized loss function *L*—drawn from *Friedman (2001)*.**

$$h = -\frac{\partial L(y_i, F(x_i))}{\partial F(x_i)} = F(x_i) - y_i.$$

## Naïve Bayes Classification (NBC)

The *e1071* package (*Therneau & Atkinson, 2015*) was used to run the NBC process using the decision factors of dimensionality, location, biomaterial, sugars, step-loading, integrins with the terminal decision variable as cell survival represented in two conditionally independent classes: short-living (i.e., apoptotic cells expected shown to undergo cell death soon after recovery) or long-living (i.e., live cells expected to survive and can be cultured and expanded successfully). Therefore, there are 6 features and 125 instances. All cryopreservation 'decision factors' are drawn from the literature cited and matched

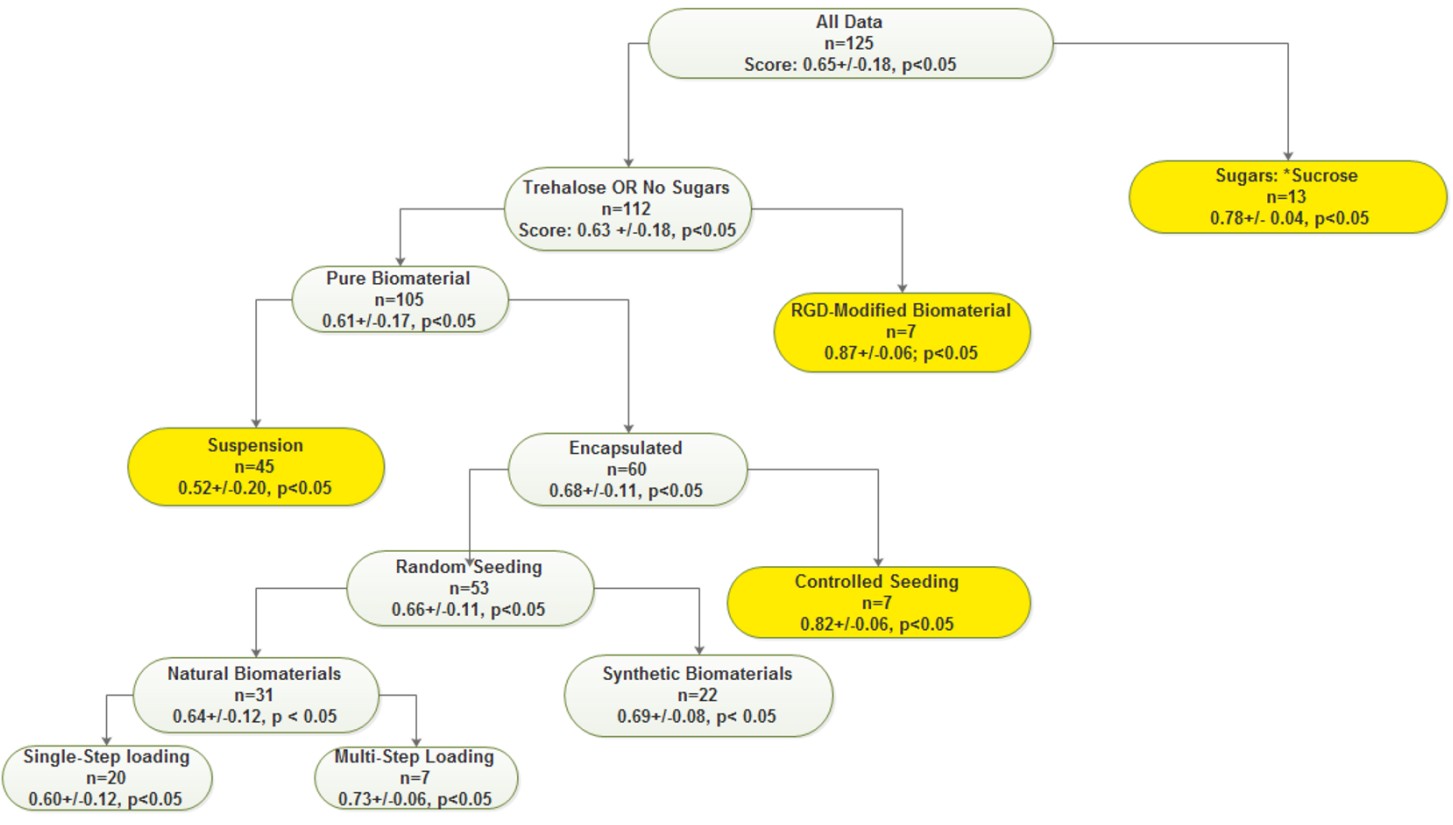

**Figure 1 The decision tree learning model.** A recursive partitioning in detail showing the hot-spots at "Integrins," "Sucrose" and "controlled seeding." The validation scores were 77% (match) and 23% (failed) for the DTL model.

semantically by author. The model is 10-fold cross-validated (giving >10 samples as test set for each model instantiation based on a >100 instances for training the model) and the model that best describes the data set is selected after evaluation for fidelity to the original data. The governing equation is provided below:-

**Equation 2: Governing equation for the NBC algorithm where *Y* is the target variable while *X* are the decision factors 161 drawn from *Mitchell (1997)*. Conditional independence enables the simpler representation within the product term.**

$$P(Y = y_k|X_i) = \frac{P(Y = y_k)\prod_{i=1}^{N}P(X_i|Y = y_k)}{\sum_{j}P(Y = y_j)\prod_{i=1}^{N}P(X_i|Y = y_j)}.$$

## RESULTS AND DISCUSSION

### A DTL analysis focussed on material and process choices

From Fig. 1, processes involving the incorporation of sugars improve survival as is captured by the terminal leaves labelled "*Trehalose*" and "*No Sugars*" with a 9% delta across both predictions coming after the introduction of a natural biomaterial. The

incorporation of sucrose into hydrophilic biomaterials modulates porosity e.g., in the alleviation of decreased porosity caused by material compression (*Huang et al., 2003*). In the context of cryopreservation, the addition of sucrose helped create further pore interconnections in the final scaffold and/or create new voids. This is primarily because of the inhibition of DMSO action by van der Waals (VWF) and hydrogen-bonding interactions between sugars, water and DMSO. These sugar-DMSO and water-DMSO interactions are expected because DMSO is prone to act as a hydrogen bond acceptor with both sugar and water molecules (*Huang et al., 2003*). The addition of sucrose as a component of the alginate capsule and as part of the cryopreservative improves the post-thaw cell viability as previous studies of such systems have shown that the effective concentration of water and the fraction of freezable water are diminished for alginate concentrations greater that 0.5% (w/w) (*Pongsawatmanit, Ikeda & Miyawaki, 1999*). This is consistent with diminished ice formation and freeze-concentration. The addition of sugars to the alginate scaffold was thought to be necessary since mass transfer barriers are eliminated and the cryoprotective benefits of sugar were expected to lower the required minimum DMSO exposure times and intracellular DMSO content (*MacGregor, 1967*). Whether in the cryopreservative or in the scaffold, the mechanism for cryoprotection involves hydrogen bonding between atoms within hydroxyl group of the sugar, DMSO and water. Sucrose can act as a hydrogen bond donor to both DMSO and water. DMSO is known to isolate water molecules and to induce hydrogen bonding (*Sato et al., 2004*). This may be related to the interaction between trehalose and alginate solutions; it has already been established that sugars impact the swelling ratios of hydrogels and by the same token, the survival of encapsulated cells (*Selmer-Olsen et al., 1999*). The higher swelling ratio of the trehalose-alginate system is linked to the lower impact trehalose has on the effective concentration of water available for alginate swelling. This will have an impact in the cryopreservation of cells since the relatively higher effective concentration of water in the alginate-trehalose system will increase the probability for ice formation which is lethal to cells. Overall, it is expected that two measurable quantities i.e., the fraction of freezable water and the effective concentration of water in alginate/trehalose scaffolds will be higher (*Martinsen, Storrø & Skjårk-Bræk, 1992*); this will in turn mean that the level of cryoprotection in these scaffolds will be lower than that observed for sucrose consistent with experimental observations and DTL analysis.

Given the differentials in cell survival for 3D capsules, the provision of optimized cryoprotection throughout the scaffold has been shown to improve overall cell survival and to minimize differentials along the construct radius (*Cui et al., 2002*); this will ensure that excessively long exposure times are eliminated from the cryopreservation process (*Sambu et al., 2011*). Comparable results indicate that cryopreservation requirements place even greater diffusion limits than do nutrient and oxygen perfusion limits (<2 mm) (*Griffith et al., 2005*); this is corroborated by the 16% (see "*Random seeding*" vs. "*Controlled Seeding*" means in Fig. 1) increase caused by the controlled seeding of cells at "location" of cells within the 3D construct for optimal nutrient, oxygen and CPA concentrations.
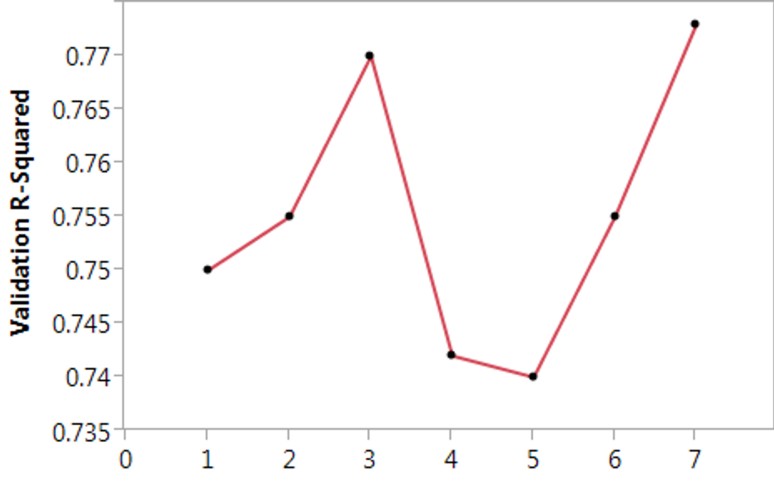

**Step (Each new branch added to Decision Tree)**

**Figure 2 Tracking the coefficient of determination as the DTL model is built.** A line plot of the R-squared for each branch added at each step in the generation of the decision tree learning model. The prediction is the cell survival (a continuous variable) after cryopreservation.

From Fig. 1, there are competitive alternatives which are found on the right-most termini of the decision tree with the use of RGD integrins and controlled seeding breaking the 80% barrier and with low $p$-values. These alternatives are optimal in their own respect; however, overall, the employment of integrins is by far the best approach.

From Fig. 2, the DTL approach is locally optimal and can often be visualized by plotting the prediction of nodes with very low $p$-values. However, there is a limit to the DTLA approach in that generalization cannot be delivered as there are no predictions outside the very local and very limited scope. As such, the proper diagnostic is the variable R-squared at each branch as shown in Fig. 2.

## An augmented DTL analysis focussed on thermal history, material and process choices

Further analysis with augmented data (*Kashuba Benson, Benson & Critser, 2008*) via the DTLA enabled further refinement of the decision process per Fig. 3 which shows that the local maximum is sustained against new process variables related to thermal history i.e., cooling rates and plunge temperatures. Given the branching mode, it is clear that process variables lead to a different decision-making path than the matrix related variables. Of note, changes on process variables lead to significant changes in the survival (see 14% increase on *Plunge Temperature* $< -60.5\,°C$). In Fig. 3, a meta-data analysis shows a survival-optimized termini with RGD-modified natural biomaterials and controlled seeding in capsules with $>80\%$ survival and low $p$-values. However, the limit in this heuristic is in generalization i.e., no predictions can be made outside the local and very limited original scope of the specific combinations within the meta-data.
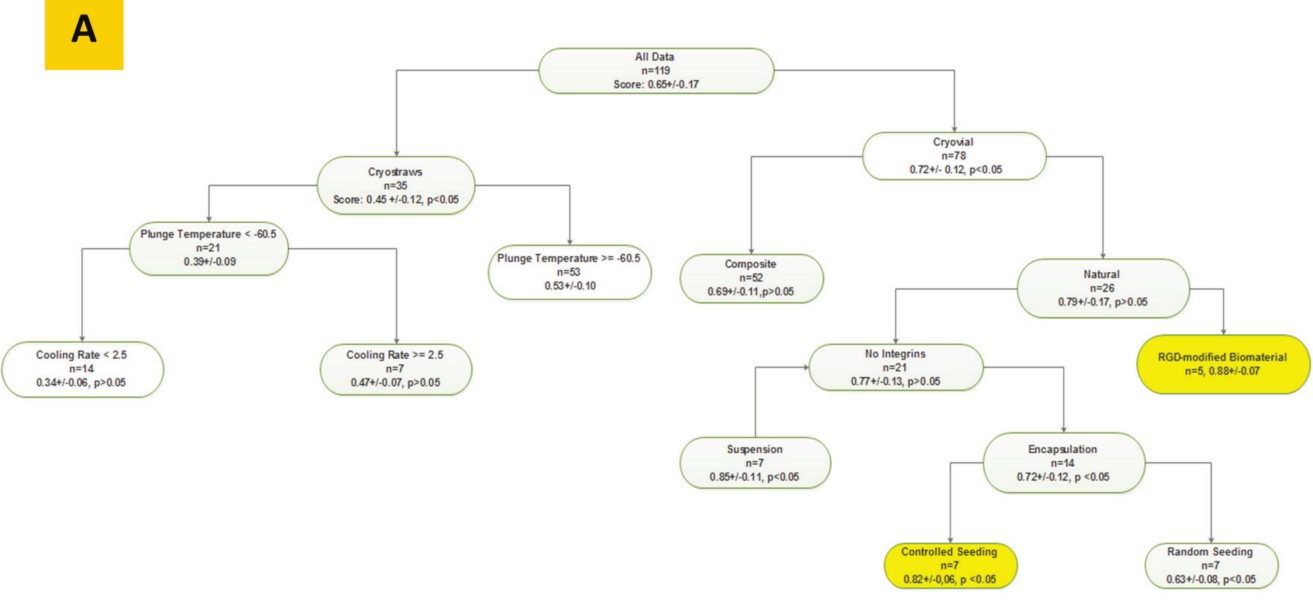

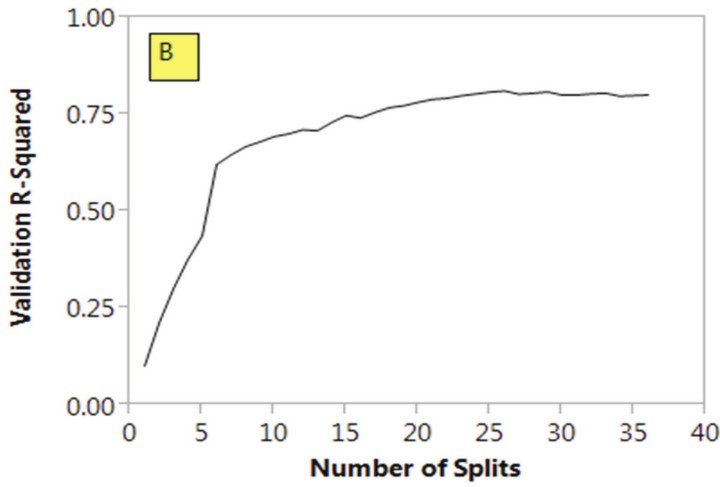

**Figure 3  A Decision Tree Learning Analysis of cryopreservation data with branch-validation tracker underneath.** (A) A metadata analysis of cryopreservation data showing that the right-side analysis is conserved across methods—Temperatures are in °C and Cooling rates in °C/min; (B) plot showing coefficient of determination during model validation; the corresponding classification scores for test data are at 84% correct match and 16% misclassification.

Before a generalization can be developed, an examination of patterns to follow in the construction of the generalizable classifier is required— Fig. 4 captures a comparison of the meta-data to be used in generating the generalizable classifier. From the graph, the variables i.e., *Step-loading*, *integrins* and *location* are particularly pivotal in the attainment of high post-thaw cell survival rates. One additional factor, i.e., *sugars*, when intersected with *location* also provided one of the 7 graphs with exceptional survival rates. In setting

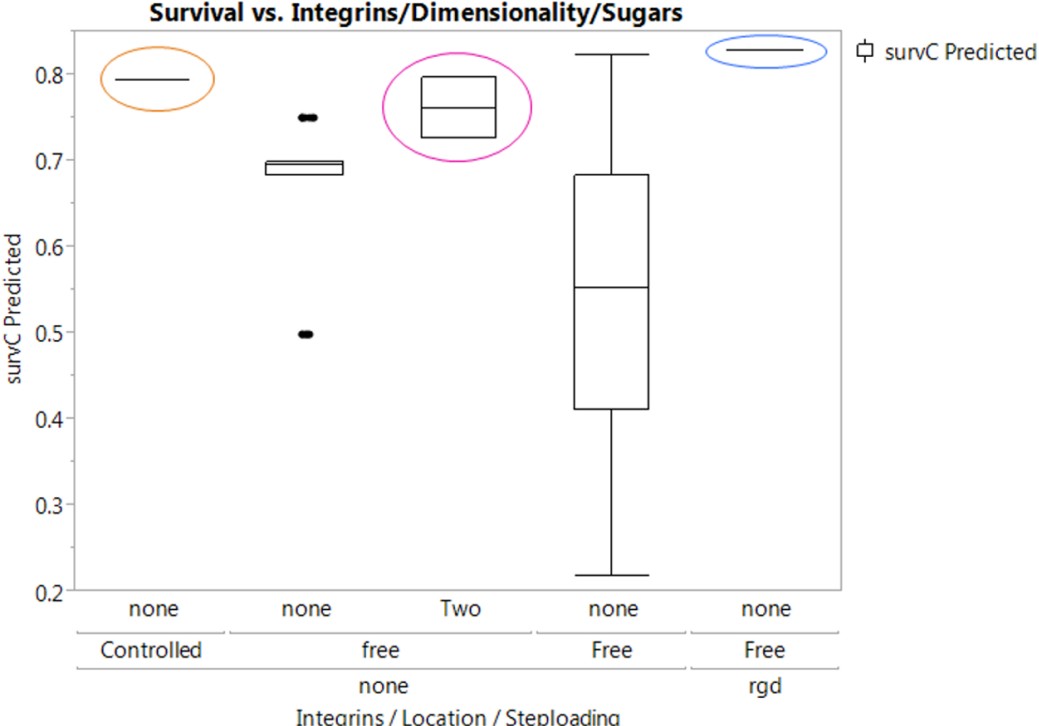

**Figure 4 A summary scatter-plot analysis of key values for meta-data on main decision factors.** A scatter plot matrix capturing the summary statistics for the meta-data collected from previous analyses used to develop a heuristic for predicting the posterior survival probabilities for cells for a given set of process decisions (*Sambu et al., 2011*; *Heng, Yu & Ng, 2004*; *Kashuba Benson, Benson & Critser, 2008*; *Miszta-Lane et al., 2007*). Key decision variable combinations with 100% "live" cells are framed in purple. The numbers on each axis represent the categories identified within a decision variable. Red dots represent short-living cells while green dots represent long-living cells.

up the meta-data sample for the generalizable NBC classifier, the selection of both robust technology-enabled cooling protocols e.g., using controlled-rate cooling and techniques using cost-efficient techniques with validated cooling rates was essential to avoid a pre-selection bias and to ensure that the predictive classifier could elicit the right probabilistic nodes from root-to-leaf. Therefore, studies e.g., by *Heng, Yu & Ng (2004)* were deliberately included with salient descriptors for protocol choices such as containment and cooling rate specified as per authors' note (these rates were verified via a thermal probe) (*Heng, Yu & Ng, 2004*). The results in latter paragraphs were examined against these data samples and the marginal descriptors introduced to capture methical improvisations.

Figure 5 shows how a change in survival against the *plunge temperature* set point significantly changes the terminal/'leaf' value. This is the temperature at which the cryovials will be immersed in liquid nitrogen for long-term storage. From Fig. 5, the plunge temperature set point is at −60 °C—while this may be applicable to the specific protocols used (*Kashuba Benson, Benson & Critser, 2008*), there may be difficulties in generalization outside of the cell type used unless the models developed are able to learn across this data set. The importance of plunge temperature can be explained using the central crybiological

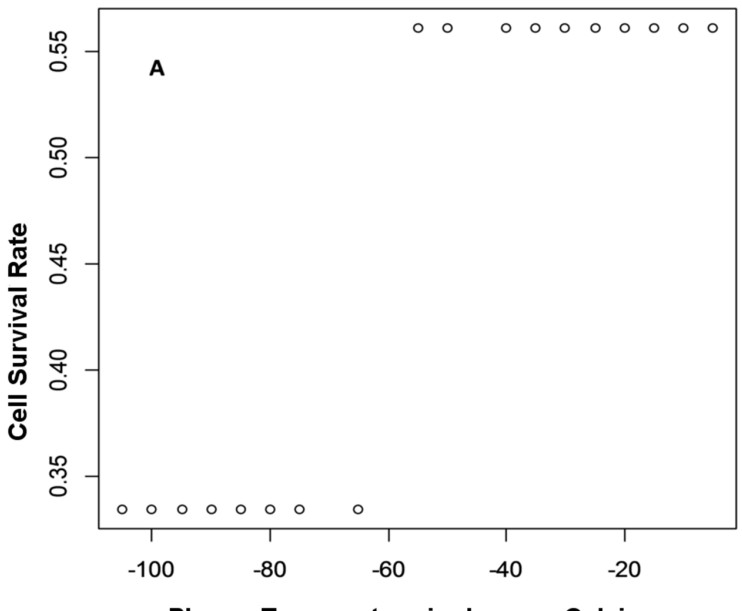

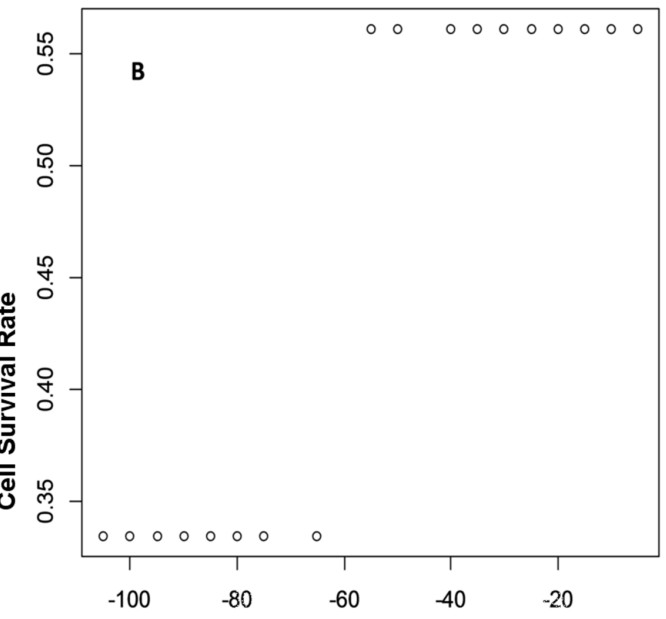

**Figure 5 A 2D plot zooming in on a classification boundary for the plunge temperature.** (A) A change in survival against the plunge temperature set point against which the prediction changes the 'leaf' value. (B) A 2D plot of the survival rate against plunge temperature was drawn from a DTL analysis. The cooling rate is used to lower the temperature from CPA loading temperature (temperature when cells are exposed to CPA for equilibration) to the plunge temperature—this rate is a separate decision factor (see methods section).

| Sample | Prediction | Experimental Long Survival % predicted Value | Short Survival % predicted Value |
|---|---|---|---|
| Test | Long Survival | 89% | 0% |
| | Short Survival | 11% | 0% |
| Train | Long Survival | 38% | 9% |
| | Short Survival | 11% | 42% |

**Figure 6 A nested summary table of the NBC model.** Prediction of survival using the Naïve Bayes Classifier on the following factors: dimensionality, location, biomaterial, sugars, step-loading & integrins. The training set has a 79% accuracy while the smaller test set has an 89% accuracy showing that the Naïve Bayes Classifier is a demonstrably accurate predictor even when reduced to a smaller test set where the samples are all cryopreserved in 3D natural RGD-containing biomaterials using controlled slow-cooling with sucrose for lyoprotection, in a two-step loading process. (Similar to the optimal leaves from the recursive partitioning trees developed earlier.) Cross-validations are 10-fold (i.e., $k = 0.1$ for the NBC model construct).

theory which states that for the cell to survive cryopreservation, intracellular ice formation (IIF) must be suppressed while intracellular vitrification must be fostered through a programmatic replacement of free water with cryoprotective chemistry—therefore, plunge temperatures that are too high (above the homogenous nucleation temperature of water, $-40\ °C$) will result in IIF while plunge temperatures that are too low will result in cell dehydration via ex-osmosis (*Meryman, 2007*).

Figure 6 shows results from the NBC method which allows generalization across the factors used whereby new combinations of otherwise conditionally independent (given its classification) decision factors can be made and evaluated for posterior probabilities. The assumption of conditional independence stems from the observation that once the classification is computed, then for any two factors, the value of one factor does little to inform the model of the state of the other factor or its contribution to the target variable (i.e., "survival"). These predictive syntheses demonstrate a generalized accuracy of 79% across the meta-data while the accuracy for predictive analytics for a specified

case whereby samples are all cryopreserved in 3D natural RGD-containing matrices, using controlled slow-cooling with sucrose for lyoprotection, in a two-step loading process shows an 89% accuracy which meets the needs for evaluation of most-likely protocol choices. The "two step-loading" qualifier refers to the process used to get the CPA into the cells whereby cells are exposed to a low CPA concentration and then a higher CPA concentration prior to the commencement of cooling to cryogenic temperatures. 2-step loading is preferred over 1-step loading because cells are less prone to damage from sudden ex-osmosis of water and equilibration is ensured prior to cooling leading to better results upon recovery (*Santis et al., 2007*). After these results were derived, an analysis of recent published and peer-reviewed data showed that loss of RGD-responsive cell attachments during cryopreservation is a leading cause of cell loss (*Terry et al., 2007*). Furthermore, there are strong indications that sucrose as a porogen helps provide *in vivo*-like conditions in a step-wise process during tissue fabrication (*Verhulsel et al., 2014*). Subsequent to the derivation of these initial results, further work has shown that expansion and cryopreservation in 3D capsule matrices for cellular aggregates is demonstrable for human embryonic stem cells further confirming that decision choices that encompass cell attachment (similar in function to RGD-containing matrices), slow-cooling, matrix permeability, natural biomaterials (e.g., alginate) enhance cell viability (*Serra et al., 2011*).

## Synthesis

A synthesis of the previous analyses shows that the use integrin ligands improve cell survival sharply—this result has been verified in more recent publications along with 4 major decision variables mentioned in the methods section (*Ahmad & Sambanis, 2013*; *Li et al., 2015*). However, where cost may be a limiting factor, the use of sugars and a careful biomaterial choice may suffice as alternatives confirmed by recursive partitioning (though only as far as survival is concerned; maintenance of cell character e.g. pluripotency may be a different matter). This result is definitively captured by the decision tree model which shows that the lead attributes for cryopreservation are: sugars, integrins, dimensionality and location according to the information-gain these attributes afford at each cycle of recursive testing.

## CONCLUSION

In conclusion, DTL and NBC were employed on cryopreservation meta-data with NBC demonstrating greater generalizability. These results indicate that the handles for improving cryopreservation outcomes are: integrin-mediated cryopreservation, the modification (by entrapment of benign CPAs or other means) of the scaffolding material and the modification of cell location in scaffolds. The DTL and NBC models were demonstrated to be robust against tougher validation data from different cell types thereby confirming that cryopreservation/bio-preservation technology can be improved upon using these approaches.

### Funding

The author declares there was no funding for this work.

### Competing Interests

The author declares there is no competing interests.

### Author Contributions

- Sammy Sambu conceived and designed the analytic approach, performed the model building, calibration and validation, wrote the paper, prepared figures and/or tables, reviewed drafts of the paper.

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
