# Peer review of "A Bayesian approach to optimizing cryopreservation protocols"

_PeerJ, doi:10.7717/peerj.1039_

## Round 0.1 · original submission · Major Revisions

The Manuscript is, in principle, quite interesting; I agree with the 3rd Referee that using machine learning to detect which conditions are most important in this problem can open new doors. Yet, I also agree with all Referees that the paper can and should be improved before publication.

Beside what you will find in the reports here below, I have some comments about the machine learning process you have used.

- First of all, in any data mining task, it is of utmost importance to perform a cross-validation. While the Manuscript states that "Thereafter, the model is cross-validated", this is not enough: which kind of cross-validation has been used? With which parameters?

- In the data collection, you should clearly specify how many instances (i.e. examples) and features (i.e. variables, please note the terminology) you have in your data set. Also, please notice that if the number of features is similar to the number of instances, the problem of the "curse of dimensionality" should be discussed. In other words: do you have enough instances to confirm that the conclusion you reached are statistically significant?

- Images should clearly be improved. For instance, in Fig. 1, you should clearly highlight which are the decision values, and which the output values of the model. Although I can identify them, please remember that the average reader of PeerJ may not be an expert in data mining!

Furthermore, Fig. 2 is not clear at all. First, DT and NBC do not predict values, but they predict classes; thus, I guess that when you say "predicted value", you refer to the fact that each value has been assigned to a class. This difference may seem trivial, but it's not, as you would then realise that representing results in a line does not make much sense. Instead, you should report the classification score, i.e. how many times the algorithm was able to correctly predict the class. But once again, this makes sense ONLY if you have more than one instance per class, and you report the results of a cross-validation.

In order to address these points, I would suggest the authors to try the following strategy. Instead of trying to predict specific values, they can divide their instances in two classes, like "have survived for a long time" and "have survived for a short time". Then, train a model and cross-validate it, and show how it is possible to really predict the class.

Reviewer 1 ·

Basic reporting

The figures are at places not scientifically accurate and need revision. For example, in Figure 2 or 3B, what are the 'original' and 'predicted' labels stand for ? One do not understand what is actually measured. The author would also gain in spending more time explaining how to read Figure 4. Finally, Figures 5A and 5B have the same description, the author should clarify the name of the axes to allow a better understanding.

Experimental design

1- It is not clear what data was collected. A more thorough description of the data set is needed to clearly identify the aim of the author. Also the last sentence of the Introduction would fit better into the material and methods section.

2- In 2.1 Data collection section, the author have used a normal distribution to redraw virtual samples ?

3- In 2.2 DTLA section: How is the cross validation performed ? This is important to explain to erase any doubt about the 100% efficiency of the classification that is shown by the R2 = 1.

4- In 3.1 section: in the list of the competitive alternatives in the neighborhoods, alternative 2 and 4 have the same name/description although they follow two separate patterns in the tree.

5 In 3.2 section, please precise what additional information is obtained in the augmented data.

Validity of the findings

The cross-validation method should be clearly explained in order to support the 100% of the classification.

Additional comments

The paper introduces the use of machine learning techniques to enhance cyropreservation. The use of a Decison Tree Learning algorithm coupled with a Naive Bayes Classification is meant to provide appropriate strategies for cell cyroprotection. I have some reservations for the implementation of these methods, as well as the clarity of both the text and the figures. Due to all this, I cannot recommend publishing the manuscript in its present form.

Reviewer 2 ·

Basic reporting

Figures are not absolutely clear. More description is needed in the legends.

Experimental design

Methods are very briefly written
The information about the source of parameters, used in the study is almost absent.

Validity of the findings

It might be very valuable to show any possible experimental outcome of predictions

Reviewer 3 ·

Basic reporting

This manuscript attempts to apply machine learning algorithms to several papers describing post-thaw survival of two different cell lines. The approach is novel and has considerable merit because, applied to the vast body of cryobiological literature, it may yield insights into unexplored pathways to improved recovery of cryopreserved tissues. The author concludes that key factors are integrin inclusion, cell position and the dimension of the construct.

This being said, the presentation of the paper as well as the depth of analysis and discussion of the results is fairly weak. In particular, I have a number of general and specific complaints about this manuscript.

General:
The introduction lacks clarity. The meaning of "Biological constructs" is unclear until later when it becomes clear that the author is talking about groups of encapsulated cells. Because this is a general audience journal, a brief introduction to the scope of the analysis as well as the analysis technique would be beneficial to draw in reluctant readers. An explanation of "decision factors" vs "distinguishing categories" would also help.

The author states that encapsulated cells will have different cell survivals at each radius without support or specificity: while this may be true, the survival of cells in clusters is dependent on factors such as cluster packing density and whether the cooling rates and concentration profiles were optimal for the interior or the exterior cells.

The governing equations would be helpful to understand exactly what is going on. For someone who uses a different programming language than "R" the specifics of the method that "R" uses or the equations themselves would go a long way to assure the reader.

Better descriptions of the decision variables should be made. For example, what does the "step loading" variable mean. Why did some authors choose it over others for mESC?

Line 168: "Given the differentials in cell survival for 3D constructs": do you mean within a construct, or between 3D constructs and single cells in suspension? Frequently in the literature, cell survival is increased by encapsulation vs non-encapsulation.

Section 3.2: My understanding of these techniques is that the effects are assumed independent. This is most certainly not the case for things like cooling rate and CPA type, among several other variables. Moreover, the cooling rate in the Heng et al study was not well controlled, and would have been much faster initially than 3-4 deg/min and slower towards -80 C. Comparing these results with other studies where constant cooling rates were enforced, either by using a controlled-rate freezer, or by using a "Mr. Frostie" controlled rate container is dangerous when, for example Kashuba et al show that 3-4 deg/min differences yield up to 7 fold improvements in post thaw survival. Also, I gather from the legend of figure 4 that an additional study was included for this section? This should be highlighted in the text.

The figures and figure legends need considerable work.


Specific:

It may be journal style, but if not, I would recommend redefining abbreviations defined in the abstract.

Figure 1: This figure needs more explanation. What does "=a" "=de" etc mean?

Figure 2, 3b: what data? what is predicted? what is the model used to create both the original and the predicted?

Figure 4: What are the numbers on the axes of these figures.

Figure 5A, 5B and text: The axes labels should be changed to reflect their meaning and/or explained in the legend. What does "suvival rate against temperature" mean? Absolute temperature? Cooling rate? Plunge temperature? What is the difference between figure 5A and figure 5B? Additionally, you do not comment in the figure legend or the text itself on how "Plunge Temperature" is not a significant variable in Kashuba-Benson et al and Kashuba et al's studies. Moreover, this plot should be discussed in the context of cryobiological theory.

Figure 6A and 6B: This figure does not display correctly for me.

Experimental design

The purpose of this manuscript is ostensibly to explore the machine-learning approach, but the selection of studies is severely limited. There are hundreds of studies using encapsulated cells, yet the authors choose only two of them. I understand that the author may have wished to limit the study to the same cell line, but this was not the case, as the manuscript by Heng et al uses a different cell line than the Kashuba-Benson and Miszta-Lane papers. Moreover, the author includes the study by Kashuba Benson but does not consider the more comprensive follow up by the same authors with four more cell lines. There should be a considerable amount of text describing study selection, variable selection, and the assumptions of the analysis.

Section 3.2: My understanding of these techniques is that the effects are assumed independent. This is most certainly not the case for things like cooling rate and CPA type, among several other variables. Moreover, the cooling rate in the Heng et al study was not well controlled, and would have been much faster initially than 3-4 deg/min and slower towards -80 C. Comparing these results with other studies where constant cooling rates were enforced, either by using a controlled-rate freezer, or by using a "Mr. Frostie" controlled rate container is dangerous when, for example Kashuba et al show that 3-4 deg/min differences yield up to 7 fold improvements in post thaw survival. Also, I gather from the legend of figure 4 that an additional study was included for this section? This should be highlighted in the text.

Validity of the findings

The discussion is not thorough in the context of the cryobiological variables considered in this manuscript. SOME of the variables are well discussed, most are not. In particular, the variables found significant by the authors of the studies upon which this one is based and their relevance to the present study are ignored. The present study is severely lacking for breadth of

Additional comments

Overall: The number of interacting protocol choices in cryobiology and their dependency on cell-to-cell variability (within cell type and, even, within donor of the same cell type) suggest that metadata analysis will require considerably more data than the present study includes. The choices made by the authors of the studies on which the present paper is based were rational in that they were used to test specific hypotheses for specific cell types.

---

## Round 0.2 · Minor Revisions

The new version of the Manuscript has substantially improved, both in content and form. I nevertheless have to ask the authors to make one last effort, in order to improve even further the quality of their paper.

Besides some comments provided by the Referee, there are a couple of issues that have to be addressed:

- Fig. 4 is indeed not simple to understand. If this is a scatter plot, showing how the different instances organise according to the values of pairs of features, shouldn't all plots have the same number of red and green points? I understand the message that this image is supposed to convey, but indeed I don't fully understand what is representing.

- All the other images should be improved. Especially, I don't like the axis labels. For instance, NTree in Fig. 3, or prM and newData$Plunge.Temperature in Fig. 5: their meaning is far from intuitive, and the reader is forced to refer to the main text for their interpretation. Also, colours could be used in Fig. 6, in order to make simpler the recognition of the different bars.

Reviewer 1 ·

Basic reporting

The authors have made a good effort to make the paper more understandable for people from other fields than biology and it seems that the outcomes of DT applied to bigger data sets can bring interesting insights.

That said, I still have some complains about the paper in its actual form.

Experimental design

With a low number of instances, why using a DT regression (line 116) instead of a classifier, more effective with data with a low number of instances.
Note that the visualization of the Rsquare vs. the number of node used in the DT is not really relevant for the paper. The important result is the classification score, which is mentioned in the figures' titles. An additional feature has been highlighted in figure 1 (suspension) and I guess that the authors wanted to highlighted in the figure 3 (second data set). Also, some p-values are not displayed in figure 1 and 3. Note also that the p-value of critical factor "controlled seeding" is superior to 0.05 in figure 1.

I actually don't understand the figure 4. Maybe it is a customary design of the journal or the field of study, but it remain difficult to understand for a reader lambda.

Validity of the findings

Concerning the other method, the fact than NBC is more effective on test data (89%) than in trained data (79%) is quite surprising (see legend figure 6). The authors would certainly want to verify the causes of such outcome.
Figure 6, in that sense, is not clear (legend and color wise) and should be revised.

Additional comments

The article in its ideas is good and the analysis of the causes of the results are clear; more effort have to be put into the visualization of those results and the clarification of the data mining steps.

---

## Round 0.3 · accepted · Accept

The author has substantially improved the Manuscript in this last version. I think that it is now much clearer, with images that provide good support to the text. It is thus time to publish the paper!